# TiO*_x_*N*_y_* Thin Film Sputtered on a Fiber Ball Lens as Saturable Absorber for Passive Q-Switched Generation of a Single-Tunable/Dual-Wavelength Er-Yb Double Clad Fiber Laser

**DOI:** 10.3390/nano10050923

**Published:** 2020-05-10

**Authors:** Ricardo I. Álvarez-Tamayo, Omar Gaspar-Ramírez, Patricia Prieto-Cortés, Manuel García-Méndez, Antonio Barcelata-Pinzón

**Affiliations:** 1Postgraduate Programs Department, Universidad Popular Autónoma del Estado de Puebla, Puebla 72410, Mexico; ricardoivan.alvarez01@upaep.mx; 2Faculty of Physics and Mathematics, Universidad Autónoma de Nuevo León, San Nicolás de los Garza 66455, Mexico; omar.gasparrm@uanl.edu.mx (O.G.-R.); patricia.prietocts@uanl.edu.mx (P.P.-C.); 3Mechatronics Division, Universidad Tecnológica de Puebla, Puebla 72300, Mexico; antonio.barcelata@utpuebla.edu.mx

**Keywords:** titanium oxynitride, DC magnetron sputtering, saturable absorber materials, Q-switched fiber lasers, fiber micro-ball lens

## Abstract

The use of titanium oxynitride (TiO*_x_*N*_y_*) thin films as a saturable absorber (SA) element for generation of passive Q-switched (PQS) laser pulses, from a linear cavity Er-Yb double-clad fiber (EYDCF) laser, is demonstrated. Additionally, the deposition of the material as a thin film covering a fiber micro-ball lens (MBL) structure is reported for the first time. The TiO*_x_*N*_y_* coating is deposited by a direct current (DC) magnetron-sputtering technique. The MBL is inserted within the laser cavity in a reflection configuration, alongside a reflecting mirror. As a result, the coated fiber MBL simultaneously acts as a SA element for PQS laser pulses generation and as an interference filter for wavelength selection and tuning of the generated laser line. Tunable single-laser emission in a wavelength range limited by dual-wavelength laser generation at 1541.96 and 1547.04 nm is obtained. PQS laser pulses with a repetition rate from 18.67 to 124.04 kHz, minimum pulse duration of 3.57 µs, maximum peak power of 0.359 W, and pulse energy of 1.28 µJ were obtained in a pump power range from 1 to 1.712 W.

## 1. Introduction

Q-switched fiber lasers are attractive coherent optical sources for generation of high energy short pulses with duration in the order of a few microseconds or nanoseconds. Their applications cover a wide range of research areas such as medical treatment and surgery, optical communications, optical sensing, material processing, remote sensing, etc. [1,2,3]. Q-switching techniques are based on the use of an optical device that is able to vary the quality factor within the laser cavity. For this purpose, Q-switched laser pulses can be obtained by active and passive methods. In this sense, unlike active Q-switched (AQS) lasers, which require an external pulse triggering signal, passive Q-switched (PQS) lasers incorporate a saturable absorber (SA) element, which modulates the losses within the laser cavity. PQS lasers generate short powerful pulses (with a repetition rate from a few to hundreds of kHz) whose characteristics are varied in a range limited by the pump power level. PQS occurs when the nonlinear absorption of the SA increases the intracavity losses and the gain medium accumulates high energy supplied by the pumping source; then, the nonlinear absorption of the SA reaches a saturation overflow leading to an energy releasing, which generates a laser pulse with energy in the order of micro-Joules. Subsequently, the process repeats when the restitution time of the SA element elapses. In order to generate PQS pulses in fiber lasers, different nanomaterials have been reported including graphene [4,5], carbon nanotubes (CNT) [6,7], metal-doped crystals [8,9], transition metal dichalcogenides (TMD) [10,11], and topological insulators (TI) [12,13]. Particularly, in the exploration of novel nanomaterials with suitable properties for optical applications, titanium nitride (TiN) has been demonstrated as an attractive plasmonic possessing nonlinear optical properties in the near-infrared (NIR) wavelength range [14]. Additionally, titanium dioxide (TiO_2_) has been demonstrated as an oxide semiconductor and insulator with nonlinear absorption for its use as a SA element in the NIR region [15]. In this regard, a number of compositions, called in general titanium oxynitrides (TiO*_x_*N*_y_*), are obtained as a result of different intermediate phases between TiN and TiO_2_. Then, TiO*_x_*N*_y_* can be achieved through nitridation of TiO_2_ or typically by oxidation of TiN. A simple method to obtain TiO*_x_*N*_y_* is by the N–O exchange during the deposition procedure from TiN targets, which leads to formation of TiO_2_ by the oxidation process [16]. Hence, different techniques including direct current (DC) reactive magnetron sputtering and vapor deposition have been used for deposition of TiO*_x_*N*_y_* films [17,18,19]. In this context, DC reactive-magnetron sputtering represents an attractive alternative method for the growth of TiO*_x_*N*_y_* films, because it allows more accurate thickness control and higher deposition uniformity compared with other thin-film deposition techniques [19]. Then, the optical properties of the TiO*_x_*N*_y_* films can be fine-tuned by the variation on gas mixture (N and O) flow during the sputtering process, the power applied to the DC source, and the working pressure [20], which makes TiO*_x_*N*_y_* suitable for optical applications in the near infra-red (NIR) wavelength range, attractive for optical fiber compatible systems.

Moreover, fiber-based interferometers have been widely studied due to their many applications in areas such as optical sensing, photonic components characterization, wavelength-division multiplexing (WDM), spectroscopy, and optical metrology, etc. [21,22,23]. In this sense, an important application of fiber interferometers lies in its use as a spectral filter for wavelength selection and tuning of generated laser lines in fiber laser configurations; for optical systems that require coherent light emission, tunable over a specific wavelength range. Different fiber-based devices have been reported for this purpose including fiber Brag gratings (FBG), multi-mode interference (MMI) filters, high-birefringence fiber optical loop mirror (Hi-Bi FOLM), tapered fiber interferometers, and in-fiber special-fiber Mach-Zehnder (MZ) interferometers, etc. [24,25,26,27,28,29,30,31,32]. In this regard, in-fiber interferometers based on a fiber micro-ball lens (MBL) exhibit suitable characteristics for its use in fiber-based optical systems. Different reported investigations studied the operation principle of a MBL interferometer (MBLI) and its use as a fiber sensor [33,34,35,36]. J. B. Eom [33] reported an optical fiber refractive index (RI) sensor by the common path interferometric system based on the use of a detachable ball lens and lucent type connector fiber patch cord. S. W. Harun et al. [34] demonstrated a simple compact glucose sensor using a MBL at the cleaved tip of a microfiber coupler and a reflector mirror. A. A. Jasim et al. [35] improved the preceding configuration for its use as a displacement sensor by measuring the reflected power of the interference spectrum as a function of the distance between the MBL and a reflective surface. M. S. Ferreira et al. [36] proposed a fiber sensor based on an array of silica microspheres. However, to the best of our knowledge, the use of a fiber MBLI as spectral filter for wavelength tuning of the generated laser lines in fiber lasers has been unexploited. In this sense, the use of a fiber MBLI as spectral filter represents a reliable option for wavelength tuning of fiber lasers due to its immunity to electromagnetic interference, high sensitivity, low cost of light coupling, fiber compatibility, ease to fabricate, and compactness. Furthermore, depositing of a material coating on the fiber MBL used as a substrate could extend the potential optical applications of fiber MBLs taking advantage of the inherent characteristics of a MBLI combined with the optical properties of the material coating.

In this paper, we demonstrate a passive Q-switched pulsed laser generation of a single-tunable/dual-wavelength Er-Yb double clad fiber (EYDCF) laser based on the use of a fiber MBL coated with a thin film of titanium oxynitride deposited by DC reactive magnetron sputtering. The spherical MBL with a diameter of 350 µm at the cleaved tip of a single-mode fiber was fabricated by the arcing technique using a fusion splicing machine. In the proposed laser configuration, the coated fiber MBL simultaneously acts as a SA element for PQS laser pulses generation due to the TiO*_x_*N*_y_* thin film, and as an interference spectral filter for single laser wavelength tuning and dual-wavelength generation.

In a pump power range from 1 to 1.712 W, the EYDCF laser generates PQS laser emission in a wavelength range from 1542.56 to 1546.2 nm and simultaneous dual-wavelength laser generation from 1541.96 to 1547.04 nm. With the maximum pump power, PQS laser pulses with a maximum repetition rate of 124.04 kHz, minimum pulse duration of 3.57 µs, maximum peak power of 0.359 W, and pulse energy of 1.28 µJ are obtained.

## 2. Fiber MBL Coated with TiO*_x_*N*_y_* Thin Film

### 2.1. MBL Fabrication and TiO_x_N_y_ Thin Film Deposition

The fiber MBL was fabricated by using a fusion splicer with special fiber processing features (Fujikura ArcMaster FSM-100M, Tokyo, Japan) and the dedicated ball lens arc fusion program of its software (AFL Fiber Processing Software FPS, ver. 1.2b, Tokyo, Japan). A segment of standard single mode fiber SMF-28 without a polymer jacket was used to fabricate the MBL structure. By using the program options of the FPS, the fiber MBL was fabricated by adjusting the ball diameter to 350 µm with minimal ellipticity. Figure 1a shows the microscope image of the fabricated fiber MBL.

The thin-film was deposited by reactive DC magnetron-sputtering of a TiN target (2” diameter, 1/8” thickness by Kurt Lesker (Jefferson Hills, PA, USA) using an Ar/N_2_ mixture at room temperature. Oxygen was supplied from the same target (surface native oxide), which was not pre-sputtering cleaned. Several depositions were previously conducted in order to ensure reproducibility of the obtained results. The MBL was mounted on glass substrates using a high vacuum tape. A photograph of this arrangement is shown in Figure 1b. As the sputtering process is energetic enough, the deposited film covered the external side of the MBL. The coated substrates were also characterized via X-Ray photoelectron spectrometry (XPS) in order to analyze the chemical properties and stoichiometry of the film covering the MBL.

The synthesis conditions employed for the nanostructured thin films are listed in Table 1. The deposition was made using a gas mixture containing argon (Ar) and nitrogen (N_2_), where Ar and N_2_ were the working and reactive gases, respectively. High purity (99.999%) Ar and N_2_ were employed for the TiO*_x_*N*_y_* deposition. The chamber was pumped down to a base pressure of 6.6 × 10^−4^ Pa before N_2_ and Ar were introduced. The flow rate of both gases was controlled during deposition by using gas flowmeters. The target-substrate distance *d* is fixed at 5 cm. A source from the DC Sputtering Ion Magnetron (Materials Science Inc., San Diego, CA, USA) was used during the deposition process. The pre-used vacuum evaporation was performed using two pumps; a mechanical JEOL75 G (Agilent, Santa Clara, CA, USA) and RP-250 Turbo Macrotorr turbomolecular V (Agilent, Santa Clara, CA, USA). The gas pressure was established using a flowmeter AERAFC-7800CD. After deposition, the thickness of the resulting film was measured ex situ by the Filmetrics equipment in reflectance mode. Measurements were conducted in regions of the substrate very close to the fiber.

### 2.2. TiO_x_N_y_ Coating Characterization

The resulting thickness of the film covering the fiber ball lens, obtained from reflectance measurements, was of about 40 nm. This thickness was chosen as a part of the experimental design, as the literature reports thicknesses in the range of 20 to 100 nm for useful optical devices, since the varying of the film thickness can increase the modulation depth, from the standpoint of shorter pulses for the Q-switched operation. Thus, further works with other thicknesses beyond 40 nm will be considered. The optical reflectance of film obtained from the UV-Vis-IR spectroscopy is displayed in Figure 2. In the figure, a peak of maximum absorbance is detected at about 510 nm. This spectrum is typical of a nitride–oxide titanium and provides a hint about the absorption of film at a low peak laser-power.

The composition of films was analyzed by XPS (K-ALPHA by Thermo Fisher Scientific, Waltham, MA, USA) using a monochromatic AlKα source of 1486.68 eV. The film surface was sputtered in situ with Ar^+^ ions in order remove the surface contamination adsorbed on samples. The dedicated software of the equipment was used to calculate the atomic concentration of elements as well as the chemical state. The binding energies were calibrated with the C1s peak at 284.5 eV. The procedure employed was carried out before and after sputtering cleaning to first get a survey spectrum, followed by high-resolution windows for Ti2p, O1s, and N1s transition spectra. A XPS analysis in depth-profile mode was also performed in order to monitor the Ti, N, and O concentrations just before the rising of the Si2p and O1s signals from glass (substrate). Figure 3 shows the XPS high-resolution spectra for (a) Ti2p, (b) O1s, and (c) N1s windows and each one showing the spectra before and after sputtering. Figure 3d includes the depth-profile curve.

A summary of the spectra binding energies (BEs) is presented in Table 2. Before sputtering, the BEs of Ti2p_3/2_ can be associated to a nonstoichiometric native oxide Ti, as the metallic state is discarded. After sputtering, that Ti2p_3/2_ signal can be associated to an oxynitride phase. About O1s, its BEs, before sputtering, can also be related to native oxide. After sputtering, the O1s can be related to an oxynitride phase. For the case of N1s, the BEs show no significant changes before and after sputtering. Thus, those signals can be related to some oxynitride phase.

A summary of BEs for titanium oxynitride related compounds found in the literature has been included in Table 3. Our value found for Ti2p_3/2_, before sputtering is comparable to that reported in [37,38]. Moreover, the tendency in BEs to become lower (from 457.98 to 453.98 eV, see Table 2), when Ti tends to nitridize, is also in agreement with published results [37,38]. For the case of N1s, our values are in close agreement with the reported ones for the oxynitride phase [20,38]. For the case of O1s, the BEs before sputtering (531.5 eV) closely agree with the reported value of native-oxide phase [38], although not being the case for the oxynitride phase. This can be attributed to the chemical state of oxygen, which can differ depending upon the compound stoichiometry (including sub-stoichiometric TiN*_x_*O*_y_* phases). With the data obtained from the XPS depth-profile analysis, it was found that our film was an oxynitride compound, with an empirical formula given by Ti_0.37_N_0.41_O_0.21_.

### 2.3. Nonlinear Optical Absorption Characterization

The characteristics of TiO*_x_*N*_y_* for its use as an absorption modulator by the saturable absorption mechanism were investigated. The nonlinear optical absorption of the sputtered TiO*_x_*N*_y_* coating was characterized by the powerscan (P-scan) technique [39]. The nonlinear measurements were conducted for excitation pulse duration of 120 fs at 1550 nm (from a 5 MHz amplified mode-locked laser source) by using the configuration described by Prieto-Cortés et al. [40]. Figure 4 shows the light transmission percentage (*T*) through the TiO*_x_*N*_y_* coating as a function of the laser incident intensity (*I*) in a range from 3.15 to 25.3 GW/cm^2^. The inset of Figure 4 shows a zoomed out view of the obtained measurements. The transmission experimental results were fitted to the hyperbolic approximated saturation model expression, given by [39]:(1)$$T=\exp\left[-L\left(\alpha_{0}+\frac{\beta I}{1+I/I_{sat}}\right)\right]$$
where *I_sat_* is the saturation intensity (intensity at the 50% of the modulation depth), *L* is the sample length, and *α*_0_ and *β* are the linear and nonlinear absorption coefficients, respectively. From the fitting procedure, the large nonlinear absorption coefficient *β* was found to be −1.998 × 10^−9^ m/W. A modulation depth of 13.7% and nonsaturable absorption of ~39% was obtained. Moreover, the imaginary part of the third-order susceptibility is responsible for nonlinear absorption effects such as saturation absorption, expressed by [41]:(2)Im(χ(3))=n02ε0cλ4πβ
where *λ* is the operation wavelength, *n*_0_ is the linear refractive index of the material, *c* is the light velocity in a vacuum, and *ε*_0_ is the electric permittivity of free space. Then, with the obtained value of *β*, the calculated imaginary part of the third-order susceptibility was of −7.498 × 10^−10^ esu.

### 2.4. Fiber Micro-Ball Lense Interfoerometer Setup and Operation Principle

As shown in Figure 5a, by using a V-Groove fiber holder, the fiber tip with the MBL was mounted on a 2-axis micrometric translation stage in front of a fixed broadband metallic flat mirror as a reflector. A micrometric screw is used to adjust and fix an initial distance between the fiber and the reflector mirror whereas the other one allows carrying out fine transverse displacements of the fiber MBL with respect to the mirror surface. The schematic of the fiber MBL configuration with transverse displacement is shown in Figure 5b. As reported by A. A. Jasim et al. [35], when input light is introduced into the fiber tip (see the fiber MBL in position 1 of Figure 5b), a portion of the input intensity *I*_0_ is reflected back to the fiber due to the fiber MBL surface (*I*_1_), whereas the part of light which passes through the MBL is reflected by the mirror and coupled back to the fiber (*I*_2_) by traveling a length difference of 2*d*_0_ in a different medium gap with refractive index *n_d_*. Then, the reflected intensity *I* exhibits an interference spectrum due to the optical path difference between both reflected optical beams in which the phase difference is given by:(3)$$\Delta \phi =\frac{4\pi }{\lambda }n_{d}d_{0}+\phi_{0}$$

Thus, the reflected intensity *I* is an interference periodical modulation which can be expressed by the two-beam interference model:(4)$$I=I_{1}+I_{2}+2{\left(I_{1}I_{2}\right)}^{1/2}\cos\Delta \phi$$

As it can be observed from the fiber MBL in position 2 of Figure 5b, when the fiber MBL is transversally displaced as a distance *L* with respect to the mirror reflector, a slight tilt angle *θ* in the mirror alignment modifies the distance between the MBL and the mirror from *d*_0_ to *d*, therefore, the free spectral range (FSR) and the wavelength position of the periodically modulated spectrum of the MBLI is modified. The phase difference is changed by a distance *d* for a transverse displacement *L*, expressed by:(5)$$d=d_{0}+L\tan\theta$$

The interference spectrum of the coated fiber MBL was characterized by using the configuration shown in Figure 6a. Input light from a fiber coupled wideband Light emitting diode (LED) source from 1400 to 1600 nm was launched into the fiber MBL through port 1 of an optical circulator (OC). The MBLI setup was connected at the OC port 2, then, the reflected interference optical spectrum was measured at the port 3 of the OC with an optical spectrum analyzer (OSA) with a resolution of 0.05 nm. Figure 6b shows the measured reflected spectrum of the fiber MBLI as a function of the transverse displacement of the fiber MBL. The measured output signal is a wavelength periodical modulation of the input signal with a wavelength period of ~5 nm defined as the FSR of the interference spectrum, given by:(6)$$FSR=\frac{\lambda^{2}}{2d}$$

A set of six transverse displacements with a 10 µm interval were applied to the fiber MBL, from the initial position. As it can be observed, micrometric transverse displacements of the MBL respecting the reflector mirror leads to a wavelength shift of the modulated interference spectrum without significant change in the FSR. By using the measured FSR, a numerical simulation of the output intensity from Equations (3)–(5) was performed, where the initial distance MBL-mirror and the mirror tilt angle were *d*_0_ = 238.24 µm and *θ* = 2.382 × 10^−4^ rad. The simulated curves are shown in Figure 6c.

## 3. PQS Tunable EYDCF Laser

### 3.1. Experimental Setup

The schematic of the EYDCF laser configuration is shown in Figure 7. The linear cavity laser includes ~1.2 m of EYDCF (CorActive DCF-EY-10/128, with the core absorption of 85 dB m^−1^ @1535 nm and inner cladding absorption of 2 dB m^−1^ @915 nm) used as a gain medium. The EYDCF is cladding pumped by a 25 W multimode high-power laser at 976 nm through a 2 × 1 + 1 beam combiner. The cavity is limited in one end by a fiber loop mirror (FLM) used as a 100% reflector. The FLM consists of a 50/50 coupler with interconnected output ports. At the other cavity end the MBLI, which consists of a fiber MBL fixed horizontally on a translation stage placed in front of a flat mirror reflector, is connected to a 10% output port of a 90/10 coupler whereas the 90% output port is used as laser power output. The unconnected ports of the 90/10 and the 50/50 couplers were isolated within a glycerin solution. At the laser output, the laser spectrum was measured with the OSA and a thermo-optical power meter was used to measure the average power and the PQS pulses were detected via a photodetector and observed by an oscilloscope.

### 3.2. Results and Discussion

The characteristics of the generated PQS laser pulses are shown in Figure 8. The measurements were obtained by using a photodetector and recorded with an oscilloscope. In order to ensure the PQS laser pulses are not related to the self-Q-switch (SQS) laser operation, previous to the PQS pulses characterization, a MBL with the same fabrication characteristics but without the TiO*_x_*N*_y_* coating was inserted within the laser cavity instead of the coated one. Then, the EYDCF length was optimized from 3 to ~1.2 m where the SQS laser emission (by inhomogeneous pump absorption of a gain fiber segment) was suppressed. Thus, we ensure that the laser pulses are generated by the nonlinear optical properties of the TiO*_x_*N*_y_* coating deposited on the fiber MBL. PQS laser pulses trains were observed from the lasing threshold reached with a pump power of 0.87 W. However, stable laser pulses were obtained in a pump power range from 1 to 1.712 W, as shown in Figure 8a. Above the maximum pump power for PQS laser operation, continuous wave (CW) laser emission was observed. With the maximum pump power for PQS laser operation of 1.712 W, laser pulses with a maximum repetition rate of 124 kHz are obtained, as observed in Figure 8b. The inset shows the pulse profile where the minimum pulse duration of 3.58 µm was obtained.

The optical spectrum of the PQS Er-Yb fiber laser is shown in Figure 9a. The measurement was obtained at the maximum pump power of 1.712 W by the OSA connected at the output port via an optical attenuation connectors configuration. A single wavelength laser emission centered at 1545.48 nm is observed. The measured −3 dB linewidth of the generated laser line is of ~0.079 nm. The optical signal-to-noise ratio (OSNR) is of ~37.7 dB. From the obtained results of Figure 8 and confirmed by the narrow linewidth optical spectrum of the laser emission, mode-locked laser operation was not observed during the laser characterization (as it can be expected from the high modulation depth of the nonlinear absorption of TiO*_x_*N*_y_*). Moreover, the laser line is generated at the wavelength at which maximum gain of the periodical modulation spectrum of the fiber MBLI is reached. Although the generated laser line forbids clearly observe the interference spectrum, the FSR of ~5 nm (measured from ~1542.5 to ~1547.4 nm) can be noticed. Figure 9b shows the average output power as a function of the pump power in the range where PQS pulses were observed. The average output power is in a range from 4.2 to 146 mW. Then, a laser output power efficiency slope of 19.8% was calculated by linear fit of the obtained results.

The characteristics of the generated PQS laser pulses as the pump power is varied are shown in Figure 10. The results were obtained with the laser settings for the measurements shown in Figure 8 and Figure 9. The obtained laser pulses exhibit a typical behavior of PQS generation. As it can be observed from Figure 10a, with the increase of the pump power, the repetition rate of the PQS pulses increases in a range from 18.67 to 124.04 kHz whereas the pulse width decreases from 11.5 to 3.57 µs. As it is shown in Figure 10b, with the increase of the pump power the estimated peak power increases from 19.56 mW to 0.359 W and the pulse energy from 0.22 to 1.28 µJ.

As discussed in Section 2.4, micrometric transverse displacements of the MBL respecting the reflector mirror leads to a wavelength shift of the MBLI modulated spectrum without a significant modification of its FSR. Moreover, as it was observed in Figure 9a, the laser line is generated at the wavelength where the maximum amplitude peak of the interference modulation spectrum is reached. Then, the generated laser line can be wavelength tuned by transverse displacements of the fiber MBL to the mirror. Figure 11 shows the wavelength tuning of the generated laser lines. As it can be observed from the laser optical spectra of Figure 11a, at the limits of a single laser wavelength tuning, the interference spectrum is shifted to a wavelength position in which two amplitude peaks reach similar maximum gain amplitudes. Then, the mode competition leads to simultaneous laser generation at the two maximum peaks. The single wavelength laser lines are tuned in a wavelength range limited by dual-wavelength laser generation at 1541.96 and 1547.04 nm whose laser lines separation coincides with the FSR of the MBLI modulation spectrum of ~5 nm. The central wavelength of the tuned laser line as a function of the micrometric displacement of the fiber MBL is depicted in Figure 11b. As it can be observed, a complete wavelength tuning period is achieved for a MBL displacement in a distance of 130 µm where dual-wavelength generation is again reached and periodically repeated. The tuning of the laser central wavelength as a function of the MBL micrometric displacement can be linearly fitted with a slope of 0.038 nm/µm.

## 4. Conclusions

In this paper, we demonstrated PQS laser pulses generation from an EYDCF linear cavity fiber laser, based on the use of the TiO*_x_*N*_y_* coated MBL fiber structure with a diameter of 350 µm. The TiO*_x_*N*_y_* thin film was deposited by the DC reactive magnetron-sputtering technique. Further, the MLB structure simultaneously acts as an interference filter for wavelength tuning of the generated laser line and for PQS laser pulses generation due to the saturation absorption of the TiO*_x_*N*_y_* coating. The tunable single laser emission in a wavelength range limited by dual-wavelength laser generation at 1541.96 and 1547.04 nm was obtained with a power efficiency of 19.8%. At the maximum pump power of 1.712 W, PQS laser pulses with a repetition rate of 124.04 kHz, maximum peak power of 0.359 W, minimum pulse duration of 3.58 µs, and pulse energy of 1.28 µJ were obtained. We demonstrated the use of TiO*_x_*N*_y_* thin films deposited onto fiber optic structures as a reliable fiber compatible alternative for PQS laser pulses generation. In addition, the novel application of a fiber MBL as a wavelength tuning interference filter and as a saturable absorber device in fiber lasers was also demonstrated.

## Figures and Tables

**Figure 1 nanomaterials-10-00923-f001:**
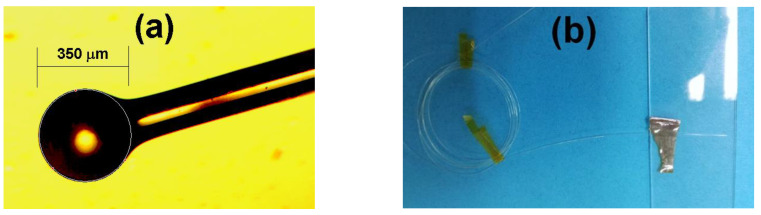
(**a**) Microscope image of the fabricated micro-ball lens (MBL), (**b**) arrangement of optical fiber mounted on glass substrate for direct current (DC) magnetron sputtering deposition of TiO*_x_*N*_y_* thin film.

**Figure 2 nanomaterials-10-00923-f002:**
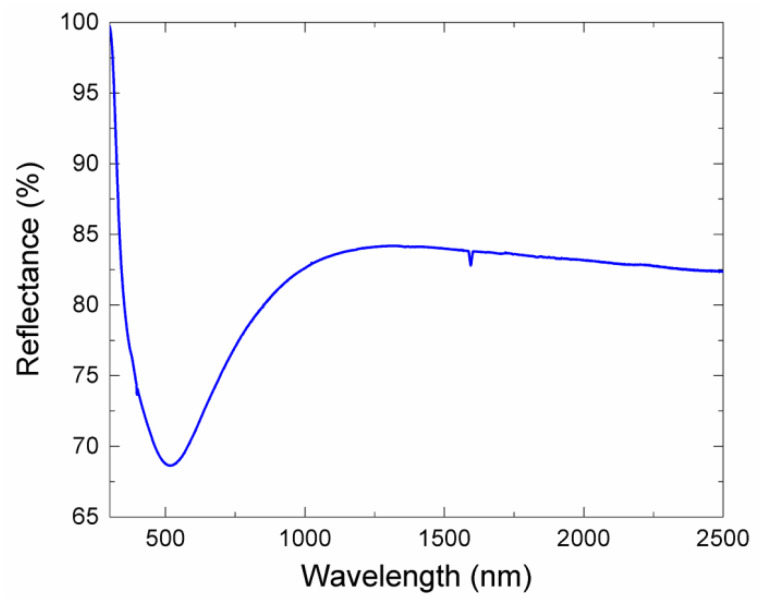
Optical reflectance of the titanium oxynitride (TiN*_x_*O*_y_*) thin film.

**Figure 3 nanomaterials-10-00923-f003:**
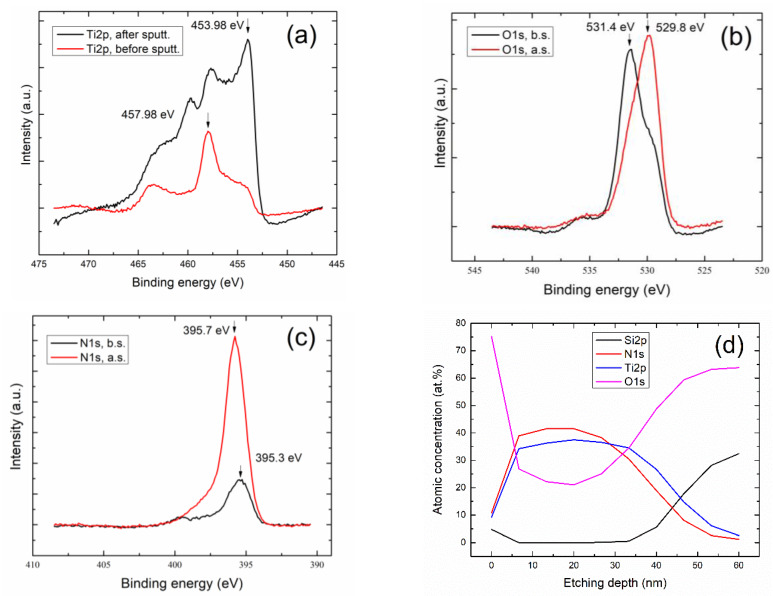
XPS high-resolution spectra for (**a**) Ti2p, (**b**) oxygen (O)1s, and (**c**) nitrogen (N)1s signals. Each one includes the spectra before and after sputtering, (**d**) XPS analysis in the depth-profile mode.

**Figure 4 nanomaterials-10-00923-f004:**
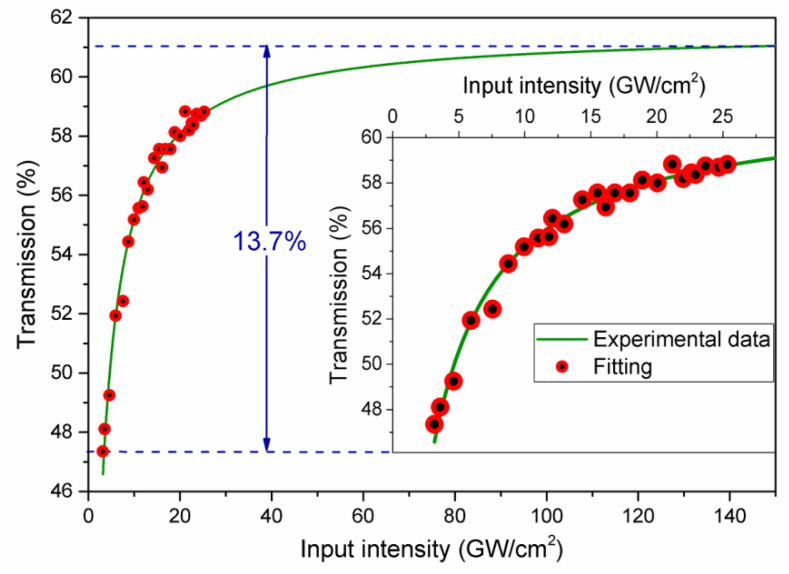
Nonlinear absorption characterization by power scan technique.

**Figure 5 nanomaterials-10-00923-f005:**
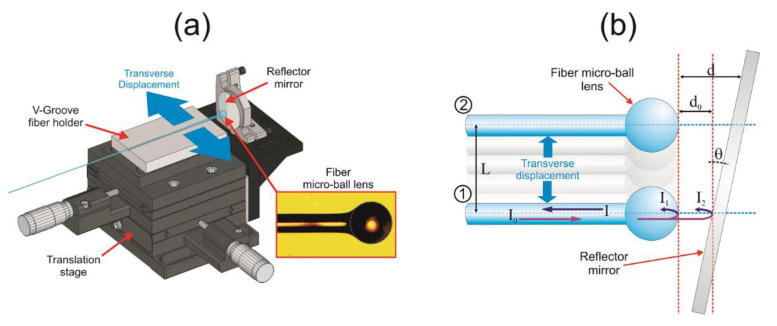
Fiber MBL implementation setup: (**a**) Schematic of the MBL interferometer (MBLI) experimental setup (inset: Microscope image of the fabricated MBL), (**b**) MBL transverse displacement schematic diagram.

**Figure 6 nanomaterials-10-00923-f006:**
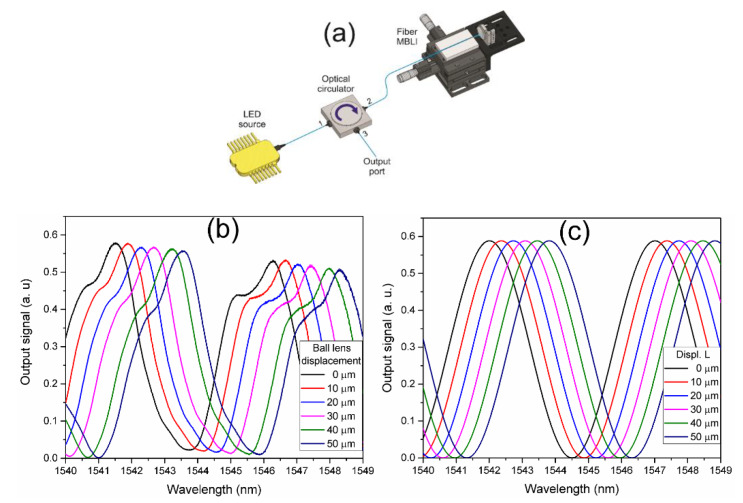
Characterization of the optical spectrum of the fiber MBLI as a function of the MBL transverse displacement: (**a**) Schematic of the used optical setup, (**b**) experimental results, (**c**) numerical simulation.

**Figure 7 nanomaterials-10-00923-f007:**
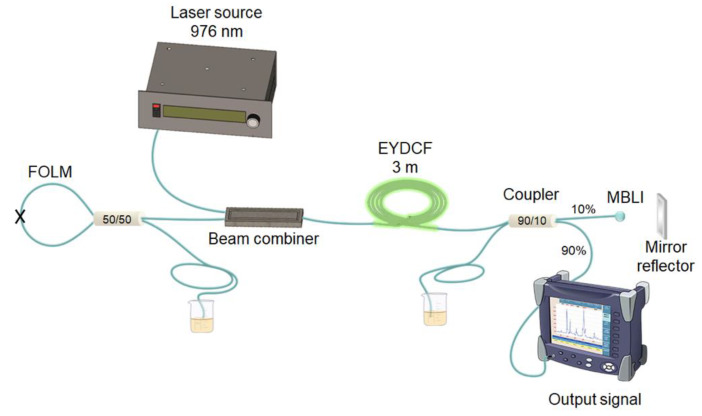
Experimental setup of the single-tunable/dual-wavelength Er-Yb double clad fiber (EYDCF) laser.

**Figure 8 nanomaterials-10-00923-f008:**
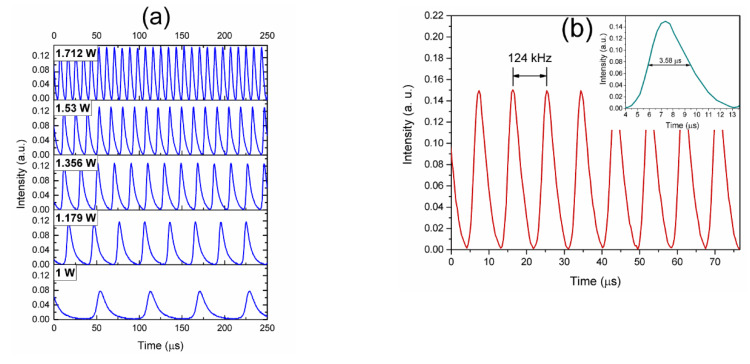
Passive Q-switched (PQS) laser pulses: (**a**) Pulse trains at different pump power levels, (**b**) pulse train repetition rate, and pulse duration at maximum pump power for PQS laser operation.

**Figure 9 nanomaterials-10-00923-f009:**
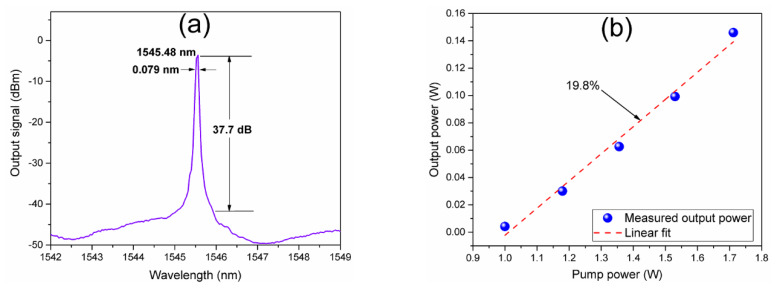
(**a**) Optical spectrum of a single laser wavelength at 1545.48 nm, (**b**) output average power as a function of the pump power in the range where PQS pulses were observed.

**Figure 10 nanomaterials-10-00923-f010:**
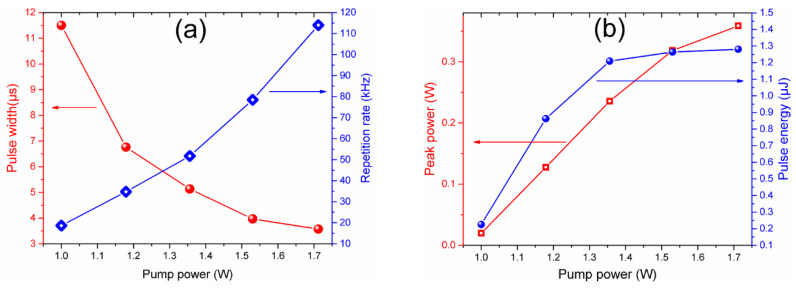
Evolution of the PQS pulses as a function of the pump power variations: (**a**) Pulse width and repetition rate, (**b**) peak power and pulse energy.

**Figure 11 nanomaterials-10-00923-f011:**
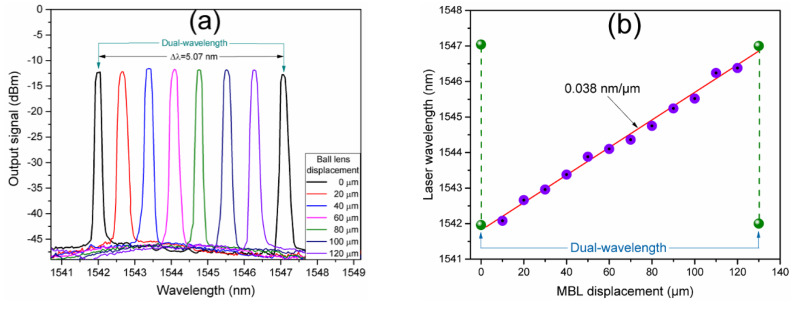
Single laser wavelength tuning and dual-wavelength laser generation by transverse MBL displacement: (**a**) Optical spectra of the generated laser lines, (**b**) central wavelength of the tuned laser line as a function of the MBL micrometric displacement.

**Table 1 nanomaterials-10-00923-t001:** Synthesis conditions of TiO*_x_*N*_y_* films deposited on optical fiber by DC reactive magnetron-sputtering.

Deposition Time	Ar Flow	N_2_ Flow	*d*	*P*	Working Pressure	Thickness
60 s	20 sccm	4 sccm	5 cm	200 W	5.0 × 10^−1^ Pa	40 nm

**Table 2 nanomaterials-10-00923-t002:** Summary of the binding energies (BEs).

Transition Signal	BEs before Sputtering (eV)	BEs after Sputtering (eV)
Ti2p_3/2_	457.98	453.98
O1s	531.4	529.8
N1s	395.3	395.7

**Table 3 nanomaterials-10-00923-t003:** Summary of BEs for titanium oxynitride related compounds found in the literature.

Transition Signal	BEs (eV)	Description	Reference
Ti2p_3/2_	458.7 455.2 455.3 459.1	-Ti2p in amorphous TiO_2_ -Ti2p in TiN -Ti2p in TiN -Ti2p in oxidized TiN	[37,38]
O1s	530 ± 0.2 531.4	-Native oxide sample -Sub-stoichiometric oxide or an oxynitride	[38]
N1s	395.7 395.1 396.0	-Intermediate oxynitride -Native oxide -Thermally bound N that is released during nitridation of Ti-O sites or in surface oxynitrides	[20,38]

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
