# Peer review of "TiOxNy Thin Film Sputtered on a Fiber Ball Lens as Saturable Absorber for Passive Q-Switched Generation of a Single-Tunable/Dual-Wavelength Er-Yb Double Clad Fiber Laser"

_nanomaterials, 2020, doi:10.3390/nano10050923_

Round 1

Reviewer 1 Report

In the manuscript submitted to the Nanomaterials the Authors describe the use of TiOxNy thin films deposited onto fiber optic structures as a fiber compatible alternative for commonly used passive Q-switched  laser pulses generator together with application of a fiber as wavelength tuning  interference filter and as saturable absorber device. The paper is very well organized, clearly written and with relatively high and new scientific content. We can find all necessary information including experimental details, description and analysis of experimental results. The manuscript can be accepted in the present form.

Author Response

"Please see the attachement"

Reviewer 2 Report

Authors have demonstrated Er/Yt doped fiber laser operation with thin film TiON-based saturable absorber in stand alone and integrated geometries for the absorber element. In the later case they have shown, for the first time, the absorber operation when it was coated on a surface of a ball lens within the fiber laser cavity. They have achieved Q-switched regime with 3 microses pulse generation at repetition rates within 20-120kHz. The output power at 1540 nm was up to 360mW at pump power levels up to 1.7W. They have also demonstrated single and dual wavelength operation by traverse displacement of the fiber with attached ball lens using a  combination of ball lens and slightly tilted reflecting mirror. The pair has effectively served as an interferometer with 6nm FSR at ~1545 nm. Very good results in the material characterization, including two-photon absorption measurements to determine the saturation power. In general, this is a well written manuscript with comprehensive characterization both on the material and laser operation sides. The work is useful for scientific community working in the fields of laser materials/modulators, high peak power lasers.

However, it would be very useful to know 1) if by varying the film thickness, the modulation depth can be increased thus  providing  better results from the standpoint of shorter pulses for the Q-switched operation. 2) what is the initial absorbtion (at low peak powers) for the thin film? 3) Did authors consider AR coating for the absorber to lower the lasing threshold. 4) And did they consider other cavity geometries for absorber-intracavity mode interaction with a goal of getting shorter pulses; 5) Any estimates/measurements  for the absorber relaxation times?  6) the Pout vs Pump slope is only 20% whereas the slope can be as high as 60-70% for Q-switch operation. What is the reason?

Author Response

"Please see the attachement"
